# The Status of Sanitation in Malawi: Is SDG6.2 Achievable?

**DOI:** 10.3390/ijerph20156528

**Published:** 2023-08-05

**Authors:** Rebekah G. K. Hinton, Christopher J. A. Macleod, Mads Troldborg, Modesta B. Kanjaye, Robert M. Kalin

**Affiliations:** 1Department of Civil and Environmental Engineering, University of Strathclyde, Glasgow G1 1XJ, UK; 2The James Hutton Institute, Craigiebuckler, Aberdeen AB15 8QH, UK; 3Ministry of Water and Sanitation, Government of Malawi, Private Bag 390, Lilongwe, Malawi

**Keywords:** sustainable development goals, sanitation, open defecation, Malawi, linear model, survey

## Abstract

Ensuring access to adequate and equitable sanitation and ending open defecation by 2030 is the focus of Sustainable Development Goal 6.2 (SDG6.2). We evaluated Malawi’s progress towards SDG 6.2 (specifically the goal to end open defecation), presenting the results of a national survey of over 200,000 sanitary facilities and evaluating their management. Based on non-linear population dynamics, we used a linear model to evaluate the reduction in open defecation between 1992–2018, and to project whether Malawi can meet the SDG target to end open defecation by 2030 under multiple scenarios of population growth. Whilst Malawi has made considerable progress in providing sanitary provision for the population, we estimate that, at the current rate of the provision of sanitary facilities, Malawi will not reach SDG 6.2 by 2030 under any of the modelled socioeconomic scenarios. Furthermore, we compare the estimates of the extent of sanitary provision classed as improved from multiple surveys, including the USAID Demographic and Health (DHS) Surveys and Government of Malawi Census data. We conclude that some of the surveys (particularly the 2015/16 DHS) may be overestimating the level of improved sanitary provision, and we hypothesize that this is due to how pit-latrines with earth/sand slabs are classed. Furthermore, we examine the long-term sustainability of pit-latrine use, investigating the challenge of pit-latrine abandonment and identifying pit-latrine filling as a cause of the abandonment in 30.2% of cases. We estimate that between 2020–2070, 31.8 (range 2.8 to 3320) million pit-latrines will be filled and abandoned, representing a major challenge for the safe management of abandoned latrines, a potential for long-term impacts on the groundwater quality, and a significant loss of investment in sanitary infrastructure. For Malawi to reach SDG 6.2, improvements are needed in both the quantity and quality of its sanitary facilities.

## 1. Introduction

Safe and accessible sanitation has been declared a fundamental human right [1]. Sanitation is central to human health, not only through disease prevention but also the promotion of human dignity and well-being [2]. A lack of safe water and sanitation is the world’s largest cause of illnesses [3]; many of these illnesses are caused by diarrheal disease, which remains the second leading cause of death in children under five, killing 525,000 children under five each year [4]. In addition to the health benefits, improvements in sanitation systems have, in some cases, been shown to have net economic benefits through a reduction in adverse health effects and health-care costs [5]. Significant steps have been made in improving access to sanitation, with over 2.1 billion people gaining access to at least basic sanitation services between 2000–2017 [6]. A lack of safe sanitation puts users at risk of faecal-oral diseases, including through exposure to contaminated drinking water. An estimated 1.8 billion people regularly use water contaminated with faeces, with 1.1 billion drinking water supplies that have an at-least moderate risk of faecal contamination [7]. Faecal contamination of water can be a source of pathogenic bacteria, viruses, protozoa, and helminths [5].

Consequently, safe sanitation is foundational to meeting several of the Sustainable Development Goals (SDGs) [2]. Globally, progress in sanitation improvements has been slow [8], with over 3.6 billion people (46% of the global population) still lacking access to safely managed sanitation and it is estimated that at the current rate of progress, 33% of the global population will still be left without safely managed sanitation by 2030 [8]. Furthermore, the number of people lacking access to improved sanitation services is only expected to grow [3]. SDG6 target 2 outlines the goal of achieving access to “adequate and equitable sanitation and hygiene for all and end open defecation by 2030” [9], with indicator SDG6.2.1.a reporting on the “proportion of the population using safely managed sanitation services” specifically [10]. Many sources report different metrics of sanitary access, including the proportion of the population with safely managed sanitation, adequate sanitation, basic sanitation, and improved sanitation [3,8,9]; this can make drawing comparisons challenging. Improved sanitation services are widely defined as ‘sanitary systems that minimize human contact with excreta including flush/pour latrines, ventilated improved pit (VIP) latrines, pit-latrines with a slab, and composting toilets’ [11], and will be defined as such for the purposes of this work. The proportion of the population with basic sanitation is defined as the percentage of the population with ‘improved sanitation facilities that are not shared with other households’ [12]. Safely managed sanitation facilities go a step further and are defined as the ‘population using an improved sanitation facility that is not shared with other households and where excreta are safely disposed of in situ or treated off site’ [13,14].

Among those lacking basic sanitation, over half live in sub-Saharan Africa [8]. Like much of sub-Saharan Africa, Malawi has a high proportion of the population without access to improved or basic sanitation. There is large variation in the estimated levels of access to sanitation in Malawi, where the percentage of the population using improved sanitation ranges between 6% [15] and 88% [16]. The 2015/16 DHS estimated that 52% of households used an improved facility [17], whilst the government’s policy document Malawi 2063 [14] estimates that 35.2% of households were using safely managed sanitation services in 2020. Malawi’s 2006 National Sanitation Policy estimated that access to improved sanitation was low, estimated as between 25–33% and dropping to less than 7% in some rural communities [18]. Malawi’s 2006 National Sanitation Policy aimed to ensure 100% of the population had access to improved sanitation by 2020; however, this target has not been met [18]. A new goal was set out in Malawi’s 2063 policy document that similarly aimed to expand sanitation services to 100% of households, but specifying a 100% use of safely managed sanitation services with a 2060 target [14]. To ensure that Malawi can meet its target [14], a revision of the 2006 National Sanitation Policy (Malawi) is needed. This revision should be informed by the status of sanitation provision, and focus on the required changes to achieve 100% coverage. 

There is further consensus on the level of open defecation in Malawi. In 2018, the Malawi Census estimated that 5.9% of the population were practicing open defecation [19], the World Bank and UNICEF also estimated that 6% of the population were practicing open defecation in 2018 [20,21]. Malawi’s government, working with NGO’s, has successfully reduced the extent of open defecation, with the percentage of the population without access to sanitary facilities falling from 27.7% [22] in 1992 to 5.9% in 2018 [19].

Population growth may undermine Malawi’s efforts to eliminate open defecation if the rate of population growth outpaces the provision of sanitary facilities; Malawi’s population is projected to increase five-fold in this century [23]. To model this non-linear growth, we chose the five Shared Socioeconomic Pathways (SSP) [24], which are outline scenarios of population growth and urbanization considering age, sex, and education [25]. Modelling the trends in sanitary provision under the current rate of growth of sanitary access and different scenarios of population growth enables projections to be made for open defecation under different socioeconomic scenarios. Projecting the level of open defecation enables an estimation of whether Malawi will meet the most basic requirement of SDG 6: ensuring access to sanitation.

To meet SDG 6—ensuring access to “safely managed” sanitation [9]—the type and management of sanitary facilities must be considered. This study explored not only the level of access to sanitary facilities, but also the type and management of the sanitary facilities. The extent of the access to improved sanitation was evaluated by investigating the type of sanitation facility through comparing the Government of Malawi Census results [19], US Aid DHS results [17,26], and the results of our extensive sanitation survey presented here. Furthermore, the survey explores the nature of sanitary management, including the disposal of non-human waste in latrines and the level of latrine collapse.

This paper evaluates Malawi’s progress in striving to reach SDG 6.2 through evaluating the access to adequate and ’improved’ sanitation services, as well as the extent of open defecation. Using an extensive, country-wide, sanitation survey, we explore the types of sanitary facilities being used and the nature of sanitary management in Malawi to address the following research questions: (1) How do our estimates of sanitary provision compare with stakeholder estimations, including the Government of Malawi, USAID, and UNICEF; (2) How are sanitary facilities managed and what are the major challenges in the management of sanitary facilities, including the disposal of non-human waste in latrines and the level of latrine collapse; (3) When, if it all, can Malawi reach open defecation-free status at the current rate of sanitary facility provision under a range of socioeconomic scenarios of population growth? These analyses provide a holistic view of whether Malawi is on track to provide safe and accessible sanitation before 2030 and whether SDG6.2 is an achievable 2030 goal.

## 2. Materials and Methods

### 2.1. Study Area

Much progress has been made in providing sanitation in Malawi, with a reduction in the percentage of the population practising open defecation, from 27.7% in 1992 [22] to 5.9% in 2018 [19]. This has involved not only the provision of sanitary facilities to the population that previously had no sanitary provision, but also providing sanitary provision for a growing population. The population of Malawi is currently almost 20 million [27] and is rapidly increasing, with an annual growth rate of 2.7% [28]. Under ‘business as usual’ population growth, modelled by the Shared Socioeconomic Pathway (SSP) Scenario 2 (SSP2), Malawi’s population is projected to reach 26.3 million by 2030 and 53.6 million by 2070 [24,25]. However, in the high population growth scenario, SSP3 projects that Malawi’s population could reach 72.1 million by 2070 [24,25]. The population growth puts increasing pressure on Malawi’s sanitation systems. Increasing urbanization also concentrates sanitary requirements, placing pressure on urban systems [14].

Providing sufficient, consistent, long-term investment into sanitation infrastructure is a challenge. Malawi is one of the least developed countries globally, classed as low, with a Human Development Index (HDI) of 0.483 in 2019, which is below the 0.513 threshold [29]; this is despite improvements in the HDI from 0.333 in 1990. Malawi is furthermore below the Sub-Saharan African average HDI value of 0.547 [29]. In addition, Malawi’s economy is particularly vulnerable to climatic shocks due to its reliance on agriculture, accounting for almost one-third of the GDP of Malawi [30], and employing over 80% of the population [31]. These factors limit a resource base for long-term investment into sanitary (and water) infrastructure.

### 2.2. Data

Data on the percentage of the population using different types of sanitary facilities were gathered from open-source reports, including the USAID Demographic Health Surveys (DHS) [17,22,26,32,33], Government of Malawi Census Data [19], and USAID Malaria Indicator Surveys (MIS) [34,35]. Data were also sourced from the UNICEF Child-related SDG Indicators [36], USAID estimates [16], and Government of Malawi estimates [14]. The number of households surveyed in the DHS, MIS, and Census is summarized in Table 1. The number of households was not provided for the UNICEF, USAID, and Government of Malawi estimates.

A survey conducted by the Government of Malawi through the Scottish Gov’t Climate Justice Fund Water Futures Programme (CJF) of 268,180 sanitation facilities was used to indicate the nature of the sanitary facilities in Malawi (Figure 1) [37,38,39,40]. The surveys were conducted by trained Government of Malawi surveyors in Chichewa and English. The responses were quality controlled by the University of Strathclyde. The data were hosted on the online platform mWater [41]. Questions were asked regarding the type of sanitary facility, usage, and the management of the sanitary facility. The types of sanitary facilities classified in each survey are summarized in Table 1. Questions were also asked about previous facilities if the facility was a replacement to a previously filled/abandoned facility. The data were cleaned to remove duplicate responses (some sites were visited more than once over time), incomplete answers, and to restrict responses to 2018–2019, resulting in 201,782 complete responses (75.2%).

### 2.3. Status of Access to Adequate Sanitation in 2018/19

An estimate of the percentage of the population with access to adequate sanitation between 2015 and 2020 was taken from a number of sources. In cases where the source listed the percentage of the population using improved sanitation, this was used as the estimate. In cases where the percentage of the population using improved sanitation was not listed, the percentage of the population using each type of sanitary facility classed as improved [11] was summed. In cases where the type of sanitary provision was divided between shared or individual sanitary facilities [22], the total number of people using each type of sanitary facility (shared and personal facilities) was summed.

The CJF Survey only investigated established sanitary facilities; therefore, the percentage of the population using each sanitary type within the DHS [17,26] and Census [15] had to be scaled to the percentage of the population with sanitary access, using each sanitary facility type (as per the specification of the SDG6.2 indicator) within the CJF Survey. Therefore, the percentage of the population using each type of sanitary facility was divided by the total percentage of the population using any of the types of sanitary facilities listed within the CJF Survey.

To estimate the percentage of the population with no access to improved sanitation, the results of the CJF Survey were scaled to account for the percentage of the population practicing open defecation. The percentage of the population using each type of sanitary facility was multiplied by the percentage of the population with access to sanitary facilities (from the 2018 Census data) [19].

To further evaluate the management of sanitary facilities, questions regarding latrine management were asked. Participants were asked what waste, other than faecal waste, was deposited in the sanitation facility. Questions were also asked about previously abandoned latrines, including why they were abandoned, how quickly it took for the previous latrine to fill up, and what was done to decommission the previous latrine. Answers were given through multiple choice.

### 2.4. Trends in Access to Sanitation

Evaluating the proportion of the population practicing open defecation provides a method to investigate the level of access to some sanitation provision in Malawi, thereby giving an indication of whether the country is on track to achieve the Sustainable Development Goals (SDGs).

All individuals not practicing open defecation were assumed to have access to a sanitary facility (improved or non-improved). The number of individuals with access to sanitary facilities per year was calculated from the product of the population for a given year and the percentage of the population not practicing open defecation. The number of individuals gaining access to sanitation provision each year was calculated from the change in the number of individuals with access to sanitary facilities. This includes people who were previously practicing open defecation and had received sanitation provision in a given year, as well as any increases in the population that have access to sanitary facilities. The rate of change in the number of people practising open defecation was calculated using Equation (1):(1)$$\frac{d(P\;OD)}{dt}=\frac{{(P}_{\left(year+t\right)}{OD}_{\left(year+t\right)})-(P_{year}{OD}_{year})}{t}$$

The percentage of the population practising open defecation for a given year was calculated using Equation (2):(2)$${OD}_{\left(year+t\right)}=\frac{\left(t\frac{d(P\;OD)}{dt}+P_{year}{OD}_{year}\right)}{P_{\left(year+t\right)}}$$
where ***P*** is the population size (in millions), ***t*** is the time-window being considered (in years), year is the year, and ***OD*** is the percentage of the population practicing open defecation.

Due to the uncertainty bounds and variation between the datasets, the trend in sanitary facility provision between 1992–2018 was calculated using a simple linear regression model, using the linear model (lm) function under the stats package in base R [44], thereby projecting a conservative estimate of the number of people with access to sanitary facilities. The estimated trend in sanitary provision is assumed to be constant. Estimates of the percentage of the population without access to basic sanitation were generated by subtracting the projected number of people with sanitary provision from the projected population of Malawi each year under multiple socioeconomic scenarios of non-linear population growth. Alternative scenarios of population growth were modelled using the 5 Shared Socioeconomic Pathway (SSP) population growth scenarios [24,25]. These outline 5 global scenarios of socioeconomic changes and provide a range of scenarios of population growth in Malawi.

The SSP scenarios under which Malawi achieved Sustainable Development Goal 6.2, to end open defecation by 2030 [9], were evaluated. Scenarios in which Malawi reaches its own development plan target of 100% of households using safely managed sanitation by 2060 [14] were also determined.

### 2.5. Projecting the Number of Abandoned Pit-Latrines

The cumulative number of abandoned pit-latrines due to filling projected over a given period was calculated using Equation (3):(3)$$A=\frac{P_{l}}{U\;F}\int_{t=year}^{year+T}{PT}_{t}\;dt$$
where ***A*** is the cumulative number of abandoned pit-latrines, ***P_l_*** is the fraction of the population with toilet access using pit-latrines, ***U*** is the number of users that share a pit-latrine, ***F*** is the time (in years) taken for a pit-latrine to fill up, year is the calendar year of the first year considered in the time span investigated, ***T*** is the length of time (in years) being considered, and ***PT*** is the population (million) with access to sanitary facilities.

The fraction of the population with access to a toilet (***PT***) using pit-latrines (***P_l_***) is estimated from the CJF Survey. The equation assumes that the percentage of people using pit-latrines does not change over time (assumed to be a valid assumption given the current lack of a fiscal resource base for enhanced sanitation investment). The population with toilet access (millions) (***PT***) is then calculated from a linear trend in sanitary provision, forecasting the number of people gaining sanitary access each year.

The number of users sharing a pit-latrine (***U***) and the time taken for a latrine to fill (***F***) are estimated from the results of the CJF Survey. An upper bound and lower bound estimation are approximated, and a weighted average is calculated. We estimated the weighted average of users and time taken to fill a pit-latrine by multiplying the median value in each range of users/ time taken to fill the latrine by the proportion of responses in this range and summing all the values. For the upper range of ‘more than 20 users’, we estimated an upper limit of 30 users (therefore an average of 25.5 users); the Sphere project recommends that one latrine should have no more than 20 users [45]. For the upper limit of the time-taken to fill the pit-latrine within the ‘more than 3 years’ response, we used an upper estimate of 20 years, based on the literature [46,47], resulting in an average of 11.5 years for the upper range.

## 3. Results

### 3.1. Results of the Access to Adequate Sanitation in 2018/19

Some facilities were measured through time; therefore, only the most recent survey was chosen for analysis, leaving 201,782 unique sanitary facility surveys selected for analysis from the CJF Survey. The sanitary facilities were grouped into the type of facility. The breakdown of the total numbers of each facility type is summarized in Figure 2 and Table 2; 24.2%, 69.3%, 88.5%, and 10.4% of sanitary facilities were classed as improved in the 2018/19 CJF Survey, 2018 Census [19], 2015 DHS [17], and 2010 DHS [26], respectively.

The CJF Survey found that most of the surveyed sanitation facilities were unimproved facilities, with 75.6% of the surveyed latrines being classed as ‘pit-latrines without a slab’. The only survey that broke down pit-latrines by the type of slab was the 2018 Census [19], which found that 83.2% of non-VIP (Ventilated, improved pit-latrine) pit-latrines with slabs had earth/sand slabs (used by 46.3% of the population), whilst 16.8% had concrete slabs (used by 9.4% of the population). This study classed all pit-latrines with a slab as improved, as was the case in the 2010 and 2015/16 DHS [18,26]. If the pit-latrines with an earth/sand slab were not classed as improved sanitary facilities, the percentage of sanitary facilities in the 2018 Census that would be classed as improved would be 19.0%.

The usage of pit-latrines was further evaluated to identify the challenges in pit-latrine management. In response to the question “Other than human waste what materials are disposed of in this sanitation facility”, 11.7% of sanitary facilities had nothing other than human waste deposited in the pit-latrine. Ashes were the most common non-human waste deposited in the pit-latrine, with 77.1% of pit-latrines having ashes deposited in them. Oil was deposited in 8.32% of pit-latrines, rubbish (including plastic bags) was deposited in 6.98% of cases, and mulching materials were deposited in 2.66% of cases.

The reasons for which a latrine was abandoned were also examined. The most common reasons latrines were abandoned were collapse from rainfall (55.7%), filling up (30.2%), and replacement by a new facility (10.7%). Other reasons included abandonment due to proximity to a water-point (1.0%), lack of money to pay a pit-emptier/ builder (1.4%), and lack of technical knowledge to build a new latrine (0.9%). Further investigation is needed to establish why the latter point would be a reason for the abandonment of a pit-latrine.

In cases where the latrine was a replacement for a previous latrine that had filled up, the participants were asked about the amount of time the previous latrine took to fill. A total of 111,377 latrines were replacements for previously filled latrines. Figure 3 and Table 3 show how long it took for these latrines to fill up, as well as the number of users. Most latrines took over 3 years to fill up (59.2%), 22.6% of latrines filled up in 2–3 years, 14.7% filled in 1–2 years, 3.0% filled in 6 months to 1 year, and 0.5% filled in less than 6 months. Overall, the survey attained responses regarding how many people used 88,395 of the previously filled latrines.

Alongside recording how long they took to fill up, 9478 surveyed sanitary facilities had responses explaining how the previous latrine was decommissioned. Overall, 58.4% of these latrines were decommissioned without any kind of decommissioning process, 34.3% were covered over, and 6.5% were filled in. In addition, 0.9% of the abandoned latrines were emptied or the waste was used for other purposes, 0.7% were mulched over and used for fertilizer, and 0.2% of latrines were emptied.

### 3.2. Trends in Access to Sanitation

To evaluate Malawi’s progress in providing sanitary provision for the population and to end open defecation, the trend in the population gaining access to sanitary facilities was evaluated. The open-source data were obtained from the US Aid Demographic Health Surveys (DHS) [17,22,26,32,33], Government of Malawi Census Data [18], and US Aid Malaria Indicator Surveys (MIS) [34,35]. Table 4 summarizes the number of people who had access to sanitary facilities as recorded in the DHS, MIS, and Census data between 1992 and 2018. The mean number of people who gained access to sanitary facilities each year is also given.

Figure 4 shows the estimated trend in the number of people given access to sanitary facilities; the confidence intervals and residuals for the trend are given in Appendix B. The historic trend is projected forward to forecast the rate at which sanitary access will be provided.

The data in Figure 5 summarize the projected percentage of the population with access to sanitary facilities until 2070 under multiple scenarios of population growth. No SSP scenarios [24,25] project that Malawi will reach 100% access to sanitary facilities (an end to open defecation) by 2030—a necessity to meet SDG 6—under the current rate of sanitary facility access and a key part of the Malawi 2063 plan [14]. Under SSP1 and SSP5, Malawi is projected to achieve an end to open defecation before 2035; SSP2 estimates that this will be reached by 2060. However, scenarios SSP3 and SSP4 project that there would be an increase in open defecation as the rate of population growth would outpace the rate of the increase in sanitation access. The uncertainty in the linear model is shown in Figure A1 Appendix B.

### 3.3. Projecting the Number of Abandoned Pit-Latrines

The number of abandoned pit-latrines projected between 2020–2070 was calculated using Equation (3). The percentage of the population with toilet access using pit-latrines was calculated from the CJF 2019 Survey, summing the percentage of the population using pit-latrines (with slab, without a slab, and VIP latrines). The percentage of the population with sanitary access utilising pit-latrines (Pl) was estimated as 99.4%.

The number of users ranged from “1–5 users” to “More than 20 users”, whilst the average time taken to fill the pit-latrine ranged from “Less than 6 months” to “More than 3 years”. For the purposes of this study, we estimate a lower bound number of users as 1 and an upper bound of 30 users; we estimate a lower bound of 6 months filling time for a pit-latrine and an upper bound of 20 years.

We estimate weighted averages of 8.03 years before the latrine filled and 6.50 users. The weighted averages are shown in Appendix C.

Between 2020–2070, there were a projected 1670 million annual toilet users in Malawi (assuming a continuous trend in the number of people with access to toilets) and 1660 million annual pit-latrine users. Taking the weighted averages for the pit-latrine fill up time and number of users, we estimate that there would be *31.8 million pit-latrines abandoned due to filling up between 2020–2070*. The calculated number of abandoned pit-latrines ranged from an upper estimate of 3320 million pit-latrines (assuming one latrine per user and the pit-latrine fills up every 6 months) to a lower bound of 2.77 million pit-latrines (assuming each pit-latrine is shared by 30 users and fills every 20 years).

## 4. Discussion

The CJF Survey results estimated that 23.0% of Malawi’s population used improved sanitary facilities (24.4% of the 201,782 sanitary facilities were categorized as improved); this was close to the 2020 UNICEF estimate [36], where 24.2% of the population were using safely managed sanitation. The estimated 23.0% of the population using improved sanitation in 2018/19 was greater than the 2010 DHS [26], where 8.8% of the population were estimated to have used improved sanitation. However, it is much less than the reported level of access to improved sanitation reported by the 2015/16 DHS [17] and 2018 Census [19], which reported that 55.1% (83.7% including shared facilities) and 63.8% of the population were using improved sanitary facilities, respectively. To meet the Malawi 2063 target of 100%, the Government should prioritize a revision of the 2006 National Sanitation Policy [18] to guide investment and to set clear metrics for implementation and management.

The most common type of improved sanitary facility in all of the surveys was pit-latrines with slabs. Slabs are covers over the pit-latrine hole that limit the opening of the pit, thereby minimizing light and insects entering the pit [48]. Slabs are a significant driver of latrine cleanliness; it is recommended that pit-latrines have concrete slabs to enable easy cleaning [48,49]. The difference in the reported level of improved sanitation is primarily observed as a difference in the ratio of pit-latrines with and without slabs. The 2018 Census [19] and 2015/16 DHS [17] reported that the majority of pit-latrines had a slab, whilst the 2010 DHS [26] and 2018/19 CJF Survey presented here found that the majority of pit-latrines had no slab. There was a reduction in the percentage of pit-latrines with a slab from the 2015/16 DHS [17] to the 2018 Census [19], which indicated an overestimation of the proportion of pit-latrines with slabs in the 2015/16 DHS. The most common form of sanitary facility in the 2018 Census [19] was a pit-latrine with an earth/sand slab, with 83.2% of non-VIP pit-latrines with slabs having an earth/sand slab. It is recommended that pit-latrine slabs are made of concrete [48]. Therefore, it is likely other surveys misidentified a ’pit-latrine with slab’ and therefore assigned non-concreted facilities as an improved facility. If these were classed as unimproved facilities, rather than improved, 19.0% of sanitary facilities in the 2018 Census [19] would have been classed as improved (rather than 69.3%); this would be closer to the CJF Survey estimate in this paper (24.4%). Some of the discrepancies in the percentage of sanitary facilities classed as improved in the different surveys may therefore be due to whether pit-latrines with basic earth/sand slabs were classified as ’pit-latrines with slabs’ or were classified as not having a slab in different surveys. The national optics of having a higher proportion of sanitary facilities classified as improved should also be considered within these discrepancies.

The 2015 DHS [17] reported the lowest estimate of open defecation, with 5.4% of the population being reported as practicing open defecation. Meanwhile, the 2018 Census [19] provided a slightly higher estimation of open defecation, reporting 5.9% of the population practicing open defecation, which is similar to the 2020 UNICEF estimation of 6.0% of the population practicing open defecation [21]. This could be linked to an increased effort to reduce open defecation around the end of the millennium development goals in 2015 [47].

The trend in access to sanitary facilities was evaluated to project the future level of access for the population under multiple SSPs [24,25]. Assuming a linear trend in the rate of the expansion and development of sanitary infrastructure, the percentage of the population (non-linear growth) with access to sanitary facilities was calculated; none of the projections of population growth predict that Malawi can meet the aim *to end open defecation by 2030*, as outlined in SDG 6 [9]. UNICEF estimates that by 2030, less than 1% of the population will be practicing open defecation if the current annual rate of reduction in open defecation continues [4,6]. This is calculated through estimating the required annual rate of reduction in open defecation and comparing this to the annual rate of reduction in open defecation [6]. Given the rate of population growth in Malawi, it will require an *ever-increasing number* of sanitary facilities to be constructed each year for Malawi *to maintain the current rate* of open defecation reduction. This paper therefore investigated the current trend in the provision of sanitary facilities, rather than open defecation reduction, enabling the rapid increase in population to be accounted for.

SSP2 models ’business as usual’ population growth [24] (Figure 5) and projects that Malawi will reach an end to open defecation (100% of the population having access to sanitary facilities) by 2060. The Government of Malawi outlined the goal of ensuring all households use ‘safely managed sanitation’ by 2060 [14]; at the current rate of development, Malawi looks likely to only end open defecation by this point, representing the minimum level of provision necessary for Malawi to meet this goal. Under the low population growth scenarios, SSP1 and 5 [24] (Figure 5), Malawi is projected to end open defecation before 2035. Meanwhile, under *high population growth scenarios*, SSP3 and 4 [24] (Figure 5), Malawi is projected to see *an increase in the level of open defecation* as the rate of the provision of sanitary facilities does not keep pace with the rate of population growth.

For Malawi to meet the international and national goals for sanitation provision, the rate of development of sanitary infrastructure will need to increase. Pit-latrines remain the primary sanitation system in Malawi, with 85.3% of the population using pit-latrines as their toilet facility [19]. We found that 99.4% of the 201,782 sanitary facilities surveyed were pit-latrines (including with/without slabs, as well as VIP latrines). Investing in the construction of pit-latrines has been critical in the Government of Malawi’s strategy to work towards achieving the millennium development goal, Target 7.C: “By 2015 to halve the proportion of people without sustainable access to safe drinking water and basic sanitation” [48] and, more recently, the sustainable development goal 6 [9], “Achieving access to adequate and equitable sanitation and hygiene for all and ending open defecation by 2030”. There has therefore been a significant increase in the population using pit-latrines in Malawi, largely driven by the reduction in people practicing open defecation. Whilst the associated reduction in open defecation has laudable benefits for public health and environmental management [48,50], pit-latrines also have the potential for negative health and environmental consequences if they are not managed effectively [51]. For example, pit-latrines can lead to both microbial and nutrient contamination of water resources [51,52,53,54,55,56]. Whilst pit-latrines provide a low-cost method of sanitation and are widely used in Malawi (and other countries), *there may be long-term consequences for Malawi’s groundwater supplies from the construction of the sheer number of pit-latrines necessary to end open defecation and service Malawi’s projected population increase*. Unless well considered and managed, the unrestrained expansion of pit-latrine construction to meet the needs of an ever-growing population may pose dangers to groundwater. There is a need to model the extent of the projected pit-latrine construction, according to the population growth patterns, to investigate contamination risks and ensure effective policy.

Another contamination risk from pit-latrines is through the deposition of non-human waste within pit-latrines. We found that 88.3% of latrines contained non-human waste. The most common non-human waste deposited in latrines was ashes (77.1%), commonly added to minimize smell [57,58]. Ash has also been suggested to have the additional benefit of minimizing groundwater contamination from pit-latrines [59]. Smell is an important consideration within the non-human waste deposited in latrines as there are reports of “disinfectants, pesticides, oil, laundry and soapy water, detergents, and car-battery acids” being added to reduce smell from latrines [58]. Indeed, *we found that 8.32% of pit-latrines had oil added*, which poses a considerable risk to the groundwater quality. The addition of this waste, rather than ash, has associated public health concerns [60]. There was also a significant proportion of pit-latrines (7.0% of cases) in which rubbish or plastic bags were deposited.

The construction of pit-latrines is not only necessary to meet the sanitary requirements of additional users (those either transitioning from open defecation or due to population growth), but also to maintain the needs of the pit-latrine-using population. This survey found that 111,377 of the 201,782 sanitary facilities were replacements for a previous latrine that had filled up. There is great variation in the literature regarding the time that pit-latrines are anticipated to last before they are filled [43,61,62,63,64,65]. We found that it typically took more than 3 years for pit-latrines to fill up; however, 40.8% of pit-latrines were found to have filled up in under 3 years. We estimated that there was an average of 6.5 pit-latrine users sharing a pit-latrine. Overall, 51.7% of the respondents reported that the latrine was used by 5 or less users, with 39.7% reporting having 6–10 users. Malawi has an average household size of 4.5, suggesting that approximately half of pit-latrines were used by 1 household only. The amount of time taken for the pit-latrine to fill did not show a statistically significant correlation with the number of pit-latrine users; this may be because pit-latrines are constructed in accordance with the number of intended users. There is significant variation in the estimates of how long pit-latrines take to fill up, with estimates varying from 3 months to over 26 years. The estimates of pits filling within a matter of months are typically cases where the pit-latrine has been constructed too small. Pit-latrines may be constructed too small either intentionally, when applying the ‘Arboloo’ method of constructing a deliberatively small latrine for use for 3 months to 2 years that is then covered in soil and a tree planted on top [66], or due to the latrine being an insufficient size for the number of users [67]. On the other hand, higher estimates vary from around 15 years [46,47] to reports of pit-latrines taking over 26 years to fill [43]. We calculated an average of 8.0 years for pit-latrines to fill up, which agrees with the average estimate of approximately 8 years provided in Brouckaert et al., 2012 [47].

Pit-latrine filling was found to be a major reason for pit-latrines being abandoned and new latrines being constructed (30.2% of cases). This is supported by findings in the literature [48,63]. We estimated that under the current rates of new latrine construction and level of pit-latrine usage and applying our estimates of the number of people who share a pit-latrine and the rate of pit-filling, *between 2020 and 2070, there would be 31.8 million pit-latrines constructed and abandoned due to filling up*. This represents a significant financial investment in sanitation infrastructure that would be abandoned, as well as presenting a challenge in providing space for the safe construction of new pit-latrines. The replacement of pit-latrines also causes delays in access to sanitation facilities whilst users find resources to build replacement latrines [18]. The concept of ‘Stranded Assets’ [37] should be considered here to guide a more sustainable sanitation investment strategy following a revision of the 2006 National Sanitation Policy [18], given that pit-latrine ‘assets’ are ultimately converted to a social, environmental, and financial liability through abandonment.

In the current ‘business as usual’ Sanitation Policy, Malawi’s government must ensure a high level of pit-latrine construction, not simply to account for the growing population and a transition away from open defecation, but also to service a sanitation system that is reliant on regular replacements. Techniques such as pit-latrine emptying have the potential to expand the lifespan of pit-latrines, thereby limiting the pit-latrine construction needed to simply replenish the existing stock [68]. Further research will be necessary to evaluate the feasibility of such techniques being economically viable solutions to this problem.

There is also a significant issue with pit-latrine collapse, with 55.7% of latrines being abandoned due to collapse and 59.2% of latrines being replacements for a previously collapsed latrine. To ensure pit-latrine longevity, and thereby further minimize the necessary replacement construction, designs to minimize the collapse of latrines, such as promoting the lining of latrines [68], should be further explored. Another challenge of pit-latrine abandonment and collapse is managing disused latrines. Best practice for decommissioning latrines stipulates that the latrine superstructure should be dismantled, the latrine should be filled and lime added to kill pathogens, and the latrine should be covered with debris piled on top [69]. However, we found that 58.4% of decommissioned latrines had no decommissioning process whatsoever, presenting a public health risk through human waste remaining exposed.

This work aimed to provide a large-scale, comprehensive overview of sanitary facility access within Malawi. The use of the CJF Survey enabled an overview of a substantial dataset of over 200,000 latrines; however, as it did not survey every sanitary facility in Malawi, some may suggest that there is a bias towards latrines that were more accessible to the surveyors. Furthermore, whilst we provide a summary of the types of sanitary facilities used across Malawi, it was beyond the scope of this study to explore the behavioural and cultural dynamics of sanitary facility usage.

Therefore, whilst we evaluate the potential access to sanitary facilities, we are unable to accurately evaluate the nature of the usage of each facility type. This is a particularly important consideration when evaluating the extent of open defecation as open defecation can still be observed when sanitary facilities are available [70,71,72].

Further work would be beneficial to explore how open defecation can be eradicated within Malawi, not only from the perspective of sanitary facility access, but also regarding community-wide cultural and behavioural change [72,73]. For the purposes of this study, we assume a linear trend into the annual growth in sanitary access and applied this alongside non-linear population growth models to estimate the number of people with sanitary facility access annually. The use of a linear model was applied given the current data on national open defecation available for consideration. However, it should be noted that the future levels of sanitary facilities access projected over this time period within this work may not follow such a model; the level of sanitary facility provision is highly influenced by multiple socioeconomic and policy factors, which would significantly impact the projected levels of open defecation summarized within this work.

Based on the findings of this work, we suggest several policy recommendations to ensure Malawi can take the necessary steps to end open defecation, which is necessary for Malawi to achieve SDG6.2.

(1)For Malawi to achieve an end to open defecation, a review and revision of the 2006 National Sanitation Policy [18] is critical to set standards, guide investment, prescribe metrics, and management targets to meet the Malawi 2063 aim of 100% coverage.(2)A revision of the 2006 National Sanitation Policy [18] should also take into account the critical need to move away from the approach of ‘Stranded Assets’ (investment in sanitation infrastructure, mainly pit-latrines, as a solution) and guide investment in sustainable and longer-term waste strategies.(3)Finally, a revision of the 2006 National Sanitation Policy should guide disruptive change in third-sector strategies, moving them from short-term solutions (pit-latrines) to longer-term sustainable sanitation investment for social, environmental, and economic good.

## 5. Conclusions

The survey presented in this paper, evaluating over 200,000 sanitary facilities, found an estimate of *only 24.2% of these facilities were classed as improved*, which is significantly lower than the 88.5% in the 2015/16 DHS [17] and the 69.3% in the 2018 Malawi Census [19]. We also evaluated Malawi’s progress in ending open defecation by projecting the rate of the provision of access to sanitary facilities alongside the projected population growth under multiple socioeconomic scenarios. *At the current rate of sanitary provision, no population growth scenario projected that Malawi will be able to meet SDG 6 and achieve an end to open defecation by 2030*. The non-linear SSP2 model, representing ’business as usual’, only projects an end to open defecation by 2060.

To meet SDG 6 under the current population growth, providing safe and accessible sanitation to all will need an ever-increasing rate of sanitary investment and provision. Furthermore, focus is needed to ensure that sanitary facilities are not just able to meet the requirements of basic sanitation, but rather, an increased quality of investment is necessary to eliminate stranded assets and ensure an increasing proportion of the population has access to improved sanitary facilities. It may also be wise to review the 2006 National Sanitation Policy [18] to also consider the risks to groundwater posed by the scale of pit-latrine-use and the resulting growth of point source human and non-human contaminant sources. Finally, there is a need for policy-set metrics to closely follow trends and for long-term modelling of sanitation requirements in order to meet the Malawi 2063 targets without stumbling into unintended consequences.

## Figures and Tables

**Figure 1 ijerph-20-06528-f001:**
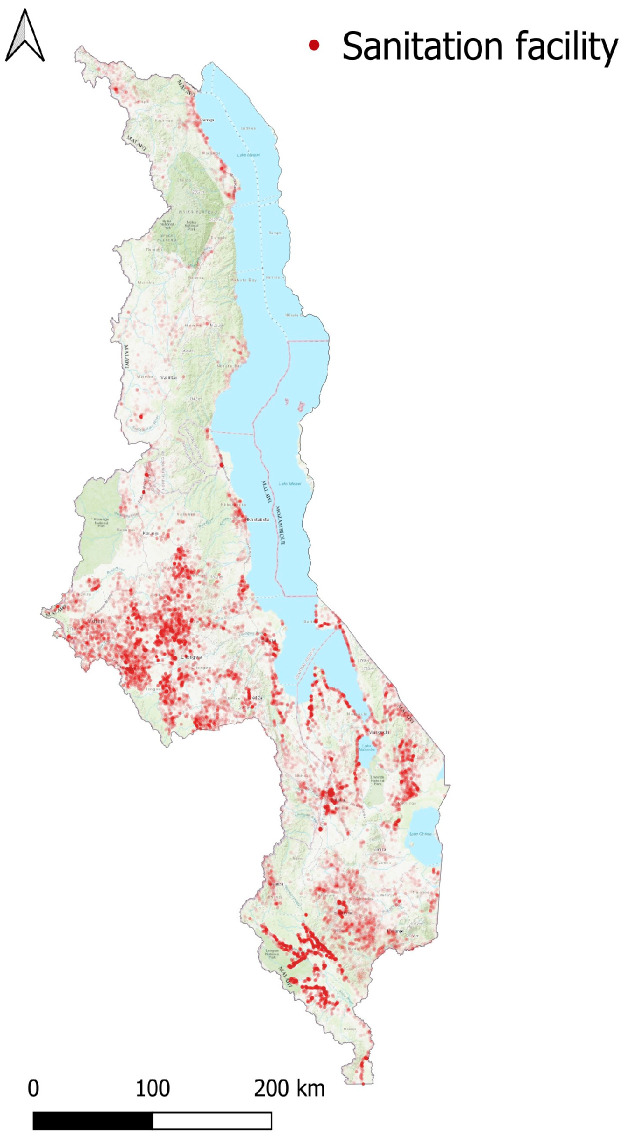
Location and distribution of the 268,180 sanitation surveys undertaken nationally across Malawi for the CJF programme. Map produced in QGIS [42]. Map background World Topo Map basemap [43].

**Figure 2 ijerph-20-06528-f002:**
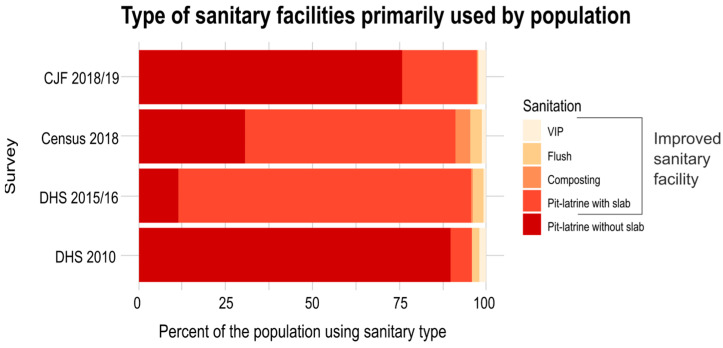
Results are grouped into 5 types of sanitary facility. The percentage of the population using each facility type includes facilities that are both shared and individual to enable comparison between surveys. Based on the type of sanitary facilities only, not accounting for whether they are shared, improved facilities are Ventilated Improved Pit-latrines, Flush latrines (including to sewer system, septic tanks, and pit-latrines), compositing toilet, and pit-latrines with slabs. Pit-latrines without slabs are classified as a non-improved facility.

**Figure 3 ijerph-20-06528-f003:**
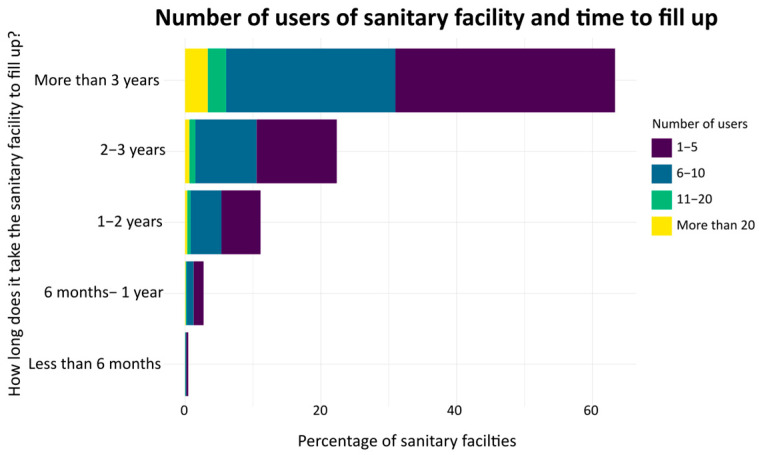
The number of people who use the sanitary facility and how long it took the previous sanitary facility (for which the current facility is a replacement) to fill up.

**Figure 4 ijerph-20-06528-f004:**
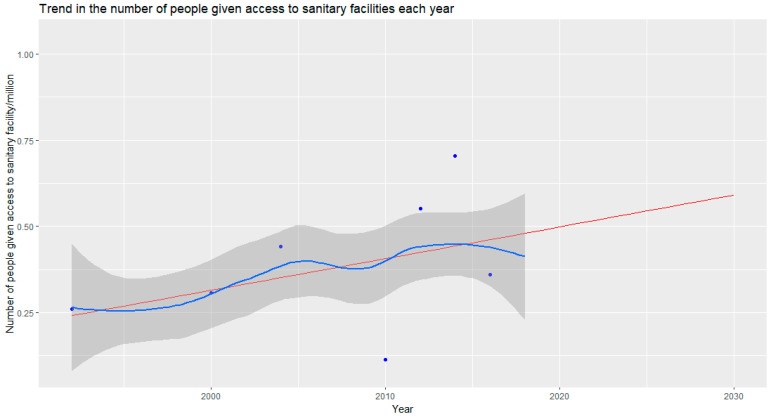
Trend in the number of people given access to sanitary facilities each year. The historical trend (data summarized in Table 4) is shown in blue. The forecast trend, through generating a linear model from the historical data, is shown in red.

**Figure 5 ijerph-20-06528-f005:**
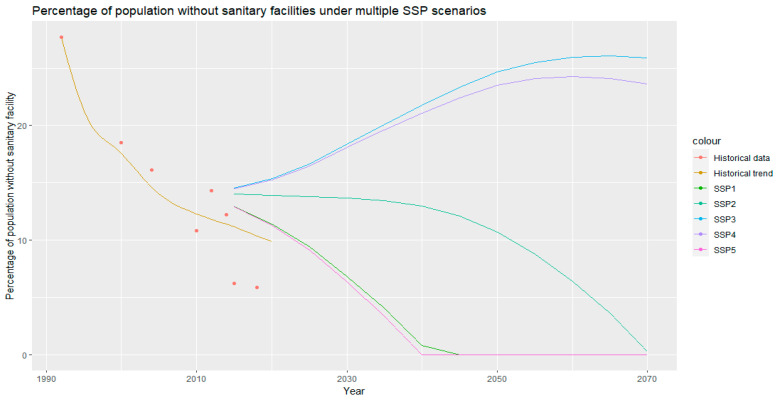
The percentage of the population without access to sanitary facilities (assumed to be practicing open defecation) under multiple socioeconomic scenarios of non-linear population growth. The historic trend (based on DHS [17,22,26,32,33], Census [19], and MIS data [34,35]) is shown alongside historical data points (from individual reports) and the projected open defecation estimates based on population growth scenarios for SSPs 1–5 [24,25].

**Table 1 ijerph-20-06528-t001:** Summary of DHS, Census, and CJF Surveys, including the number of households/sanitary facilities surveyed and the classifications of sanitation facilities in the survey. Sanitary facilities that were classed as improved for the purposes of this study are marked with an asterisk (*).

Source	Year	Number of Households/Sanitation Facilities Surveyed	Types of Sanitation Facilities Classified
DHS	2010	24,825	Flush/pour flush to piped sewer system *; Ventilated improved pit (VIP) latrine *; Pit-latrine with slab *; Any facility shared with other households; Pit-latrine without slab/open pit; No facility/bush/field; Other; Missing
DHS	2015/16	24,721	Flush/pour flush to: piped sewer system *, septic tank *, pit-latrine *; Ventilated improved pit (VIP) latrine *; Pit-latrine with slab *; Composting toilet *; Shared facilities (that would be considered improved if they were not shared by two or more households); Pit-latrine without slab/open pit; Bucket; No facility/bush/field
Census	2018	3,984,981	Flush toilet *; Ventilated improved pit (VIP) latrine *; Pit-latrine with concrete slab *; Pit-latrine with earth/sand slab *; Composite toilet *; Pit-latrine without slab/open pit; No facility/bush/field;
CJF survey	2018/19	201,782	Flush/Pour flush toilet *; Ventilated improved pit (VIP) latrine *; Pit-latrine with slab *; Composting toilet *; Hanging toilet; Hanging latrine; Pit-latrine without slab/open pit; Bucket; Other;

**Table 2 ijerph-20-06528-t002:** Italics are used where the percentage of the population referred to includes shared facilities. An asterisk (*) marks cases where the percentage of the population includes cases where sanitary facilities are classed as ‘other’ or missing values. Bold indicates figures where the result has been calculated by scaling estimates between the percentage of the population and the percentage of sanitary facilities.

Source	Year	Population Using Improved Sanitation (%)	Population Practising OD (%)	Households Using Improved Sanitation (%)	Sanitary Facilities Classed as Improved (%)
US Aid	2022	6			
UNICEF	2020	24.2	6.0%		**25.7**
DHS	2010	8.8	9.9 (* *15.4*)	8.2	**10.4**
DHS	2015/16	55.1 (*83.7*)	5.4	51.8 (*83.0*)	**88.5**
Census	2018	63.8	5.9 (* 7.8)	62.3	**69.3**
GOM Malawi 2063	2020			35.2	
CJF survey	2018/19	**23.0**			24.2

**Table 3 ijerph-20-06528-t003:** Number of users using the latrine and the time it takes a pit-latrine to fill up. On average, 51.7% of pit-latrines had 1–5 users, 39.7% had 6–10 users, 4.2% had 11–20 users, and 4.4% had more than 20 users. The distribution of the number of latrine users within each group of filling up rate was not statistically significantly different from the overall percentage of the number of latrine users for all latrine filling rates. The data values and ANOVA results are summarized in the Appendix A, Table A1.

Time to Fill Up	1–5 Users (%)	6–10 Users (%)	11–20 Users (%)	More Than 20 Users (%)
More than 3 years	51.1	39.4	4.2	5.3
2–3 years	52.9	40.3	3.95	2.85
1–2 years	51.9	40.9	4.33	2.84
6 months–1 year	54.2	37.4	5.22	3.15
Under 6 months	56.3	36.6	4.02	3.07

**Table 4 ijerph-20-06528-t004:** Change in the population of Malawi and the number of people with access to sanitary facilities from US Aid Demographic Health Surveys (DHS) [17,22,26,32,33], Government of Malawi Census Data [19], and US Aid Malaria Indicator Surveys (MIS) [34,35] from 1992 to 2018.

Source	Year	Population/Million	Number of Households Surveyed	Percentage of The Population Practising Open Defecation	Number of People Practising Open Defecation/Million	Mean Annual Increase in Sanitation Access/Million
DHS	1992	9.69	5323	27.7	2.68	0.261
DHS	2000	11.2	14,213	18.5	2.06	0.308
DHS	2004	12.3	15,041	16.1	1.98	0.442
DHS	2010	14.5	24,825	10.8	1.57	0.114
MIS	2012	15.4	3500	14.3	2.20	0.552
MIS	2014	16.3	3501	12.2	1.99	0.704
DHS	2015/16	16.8	24,721	6.2	1.04	0.386
Census	2018	17.5	3,984,981	5.9	1.03	

## Data Availability

Confidential data were obtained from the Government of Malawi. Additional data were obtained from publicly surveys including US Aid Demographic Health Surveys (DHS) Government of Malawi Census Data and US Aid Malaria Indicator Surveys (MIS).

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
