# Peer review of "The Status of Sanitation in Malawi: Is SDG6.2 Achievable?"

_ijerph, 2023, doi:10.3390/ijerph20156528_

Round 1
Reviewer 1 Report
The main content of this article is about the health conditions in Malawi. The author evaluates Malawi's progress in achieving Sustainable Development Goal 6.2 (SDG6.2), provides national survey results of over 200,000 health facilities, and assesses their management. The author constructs a linear model to evaluate the reduction of open defecation from 1992 to 2018 and assesses whether Malawi can achieve the SDG target before 2030 under various population growth scenarios. Although Malawi has made considerable progress in providing health facilities for the population, the author estimates that Malawi will not be able to achieve SDG 6.2 by 2030 under any simulated socio-economic conditions based on the current rate of providing health facilities. The author concludes that some surveys (especially the DHS 2015/16 survey) may overestimate the level of improvement in health facilities and assume that this is due to how pit latrines with soil/sand slabs are classified as improved or unimproved. Additionally, the author studies the long-term sustainability of pit latrine use, investigates the challenges of pit latrine abandonment, and identifies that filling pit latrines accounts for 30.2% of cases of abandonment. The author estimates that between 2020 and 2070, 31.8 (range 2.77-3320) million pit latrines will be filled and abandoned, posing significant challenges for the safe management of abandoned toilets, long-term impacts on groundwater quality, and significant losses in investments in sanitation infrastructure. To achieve SDG 6.2, improvement in both quantity and quality of health facilities is necessary. Overall, I think this article meets the publication standards of this journal with a clear theme and well-structured argument. However, to improve the article, the author could focus on the following areas.
(1) In terms of the article format, the author should double-check the consistency of proper nouns throughout the article, ensure that the grammar is smooth in some sentences, and conform to the journal's publication standards regarding the formatting of tables, figures, and references.
(2) In terms of the overall content of the article, the author discusses the level of health facilities in Malawi and explores the types and management of health facilities. However, there is no detailed analysis of the reasons behind this situation. It is suggested that the author add this content in appropriate places in the article or use a simple mind map to logically express it.
(3) In terms of the research methods used in the article, the author uses basic and easy-to-understand methods to calculate data related to health facilities. The author also uses a combination of normative economics and empirical economics. However, it should be noted that although linear models are easy to explain certain trending issues, they may not perform as well as other more complex models for 1) nonlinear problems and 2) high-dimensional datasets. It is suggested that the author provide more explanations for the calculation of the number of people who have access to health facilities each year, such as how this data was obtained, the validity of the data, and whether there are relevant literature to support the model's predictions, so that readers can better read and think actively.
(4) In terms of the discussion section of the article, compared with existing research, where is the marginal contribution of this study? It is suggested that the author further condense and clarify this point while also supplementing the limitations of this study.
(5) In terms of the conclusion of the article, the recommendations are too general. It is suggested that the author retain the conclusion and combine the recommendations and prospects into another section. The goal of SDG6.2 is to achieve access to adequate and equitable sanitation and eliminate open defecation by 2030. To achieve this goal, the quantity and quality of health facilities must be improved, and the management of health facilities should also be improved. It is recommended that this section can be supplemented with measures in a point-by-point manner and related stakeholders, which will appear more practical and operable.
Overall, I like the research topic and logical structure of this article. I suggest that the author make appropriate modifications and resubmit for publication. Good luck!
The main content of this article is about the health conditions in Malawi. The author evaluates Malawi's progress in achieving Sustainable Development Goal 6.2 (SDG6.2), provides national survey results of over 200,000 health facilities, and assesses their management. The author constructs a linear model to evaluate the reduction of open defecation from 1992 to 2018 and assesses whether Malawi can achieve the SDG target before 2030 under various population growth scenarios. Although Malawi has made considerable progress in providing health facilities for the population, the author estimates that Malawi will not be able to achieve SDG 6.2 by 2030 under any simulated socio-economic conditions based on the current rate of providing health facilities. The author concludes that some surveys (especially the DHS 2015/16 survey) may overestimate the level of improvement in health facilities and assume that this is due to how pit latrines with soil/sand slabs are classified as improved or unimproved. Additionally, the author studies the long-term sustainability of pit latrine use, investigates the challenges of pit latrine abandonment, and identifies that filling pit latrines accounts for 30.2% of cases of abandonment. The author estimates that between 2020 and 2070, 31.8 (range 2.77-3320) million pit latrines will be filled and abandoned, posing significant challenges for the safe management of abandoned toilets, long-term impacts on groundwater quality, and significant losses in investments in sanitation infrastructure. To achieve SDG 6.2, improvement in both quantity and quality of health facilities is necessary. Overall, I think this article meets the publication standards of this journal with a clear theme and well-structured argument. However, to improve the article, the author could focus on the following areas.
(1) In terms of the article format, the author should double-check the consistency of proper nouns throughout the article, ensure that the grammar is smooth in some sentences, and conform to the journal's publication standards regarding the formatting of tables, figures, and references.
(2) In terms of the overall content of the article, the author discusses the level of health facilities in Malawi and explores the types and management of health facilities. However, there is no detailed analysis of the reasons behind this situation. It is suggested that the author add this content in appropriate places in the article or use a simple mind map to logically express it.
(3) In terms of the research methods used in the article, the author uses basic and easy-to-understand methods to calculate data related to health facilities. The author also uses a combination of normative economics and empirical economics. However, it should be noted that although linear models are easy to explain certain trending issues, they may not perform as well as other more complex models for 1) nonlinear problems and 2) high-dimensional datasets. It is suggested that the author provide more explanations for the calculation of the number of people who have access to health facilities each year, such as how this data was obtained, the validity of the data, and whether there are relevant literature to support the model's predictions, so that readers can better read and think actively.
(4) In terms of the discussion section of the article, compared with existing research, where is the marginal contribution of this study? It is suggested that the author further condense and clarify this point while also supplementing the limitations of this study.
(5) In terms of the conclusion of the article, the recommendations are too general. It is suggested that the author retain the conclusion and combine the recommendations and prospects into another section. The goal of SDG6.2 is to achieve access to adequate and equitable sanitation and eliminate open defecation by 2030. To achieve this goal, the quantity and quality of health facilities must be improved, and the management of health facilities should also be improved. It is recommended that this section can be supplemented with measures in a point-by-point manner and related stakeholders, which will appear more practical and operable.
Overall, I like the research topic and logical structure of this article. I suggest that the author make appropriate modifications and resubmit for publication. Good luck!
Author Response
Thank you very much for taking the time to review the article and to provide such supportive and helpful feedback, it is much appreciated.
Regarding your comments and suggestions.
Comment 2) As you noted, we do not discuss the reasons behind the types of health and sanitary facilities within Malawi but rather only offer comment on the numbers of sanitary facilities. Whilst this would be a helpful addition to the paper, it was beyond the scope of this study to discuss the specific nature and use of sanitary facilities in more detail. We are currently conducting further work at a more local level to provide greater insight into the nature of sanitary facilities and their usage in two districts in Malawi, however, this work implements an alternative study design and study area therefore did not feel to be a cohesive fit within this paper.
Comment 3) It is well noted that linear models certainly have their faults when it comes to more complex trends. In this study population growth was modelled as a non-linear trend and we applied a linear model to estimate the trend in the number of people gaining access to sanitary facilities (which was then compared to non-linear models of population growth to provide an estimate of the percentage of people without sanitary access). As a complex issue influenced by many socioeconomic factors, we recognise that this may indeed not be a linear trend but it is methodologically acceptable to apply within the uncertainty bounds of the various datasets. Furthermore, as an issue which is significantly influenced by policy and governance issues we also accept that any current trends are likely to change significantly under alternate policy scenarios and we recommend new Policy sets metrics that allow better future evaluation and modelling.
We felt that we had a number of options regarding modelling future trends in sanitary facilities. The simplest analysis would be to set a constant number of people gaining access to sanitary facilities each year based on an average over the approximately 30-year period we evaluate. However, our data showed that the number of people gaining access to sanitary facilities each year was increasing therefore applying a constant number of people gaining access to sanitary facilities annually would be an oversimplification and not account for the significant investment and growth in sanitary infrastructure over the study period. We therefore wanted to evaluate the trend in the growth of sanitary infrastructure.
We recognise that the actual trend in the number of people gaining access to sanitary facilities each year could take many forms. For the purposes of this study we felt that we did not have sufficient data to substantiate any trend other than a linear trend in the number of people being given sanitary access and therefore we have modelled the growth as linear.
Comments 4 and 5) We agree that a more detailed policy-focussed framing would help to make the message clearer to stakeholders. We have therefore added an additional policy-focused (see revised Section 4 Discussion and Conclusions). We have particularly focused on proposing revisions to the 2006 Malawi Sanitation Policy (which we have added more detail regarding within the introduction too) as this was highlighted in personal correspondence with stakeholders.
We have also revised the grammar with particular attention paid to proper nouns.
We sincerely hope that they changes sufficiently address your concerns. Thank you again for your comments and feedback.
Reviewer 2 Report
This paper presents a prospective study, acording to present and past data, to achieve almost zero apen defecaton in Malawi. For this, data for different good sources were collected and analyzed, and a good linear prediction under uncertainty was determined.
In my opinion, it is a relevant and well-founded study, which can be pattern on what the political, social and economic initaitives have been achieved on the topic.
However, I am not aware that a linear prediction can give accurate stimated information, since the population trends in growing, for example, are not linear. Thus, it would be relevant some validation data with other similar studies where linear regression is used regarding population behaviour trends.
Author Response
Thank you very much for taking the time to read and review the paper, your highly supportive comments are greatly appreciated.
It is well noted that linear trends have very limited benefit in applying to population projections and great uncertainty exists within evaluating future population patterns. However, please note that population dynamics were non-linear, and we only applied a linear change to the number of people gaining access to sanitary facilities through time and then combined this with non-linear population dynamics under the 5 shared socioeconomic pathway (SSP) scenarios to look at how the number of people without sanitary facilities changed under population growth scenarios.
When evaluating the number of people gaining access to sanitary facilities annually, one option was to apply a constant number of people gaining access to sanitary facilities annually. However, we noted that this would be too reductive and not account for the improvement in the number of people annually gaining sanitary facilities that we noted over the 30-year period. We therefore evaluated the trend in the access to sanitary facilities. With the available data, we felt that a linear model was the best substantiated to estimate the current trend in the rate of access to sanitary facilities. It is well noted that this is indeed a limitation of the study and this has been emphasised further in the discussion. Furthermore, the methodology of using non-linear population models alongside the linear model of sanitary access has been emphasised throughout the paper.
We hope that this clarifies any concerns surrounding the choice of linear modelling and thank you again for your comments and feedback.
Reviewer 3 Report
This study highlights the findings of an evaluation of Malawi's progress towards SDG 6.2. The study examined the provision of sanitary facilities through a national survey. A linear model was used to assess the reduction of open defecation and the country's ability to achieve the SDG target by 2030. The study suggests that Malawi may not reach SDG 6.2 by 2030 based on current rates of provision. Additionally, the accuracy of surveys estimating improved sanitary provision is questioned, and the challenge of pit-latrine abandonment and filling is identified as a significant issue for long-term sustainability. The study concludes that improvements in sanitary facilities are needed for Malawi to achieve SDG 6.2.
In line 37-39, the authors stated that the lack of safe water and sanitation is the world's largest cause of illnesses and the second leading cause of preventable childhood deaths. They further mention that around 4,100 children die daily from waterborne diseases. It seems that the statement refers to the global figure, not specifically to Malawi. Are children under the age of five 5? I would advise to double check the figure.
In lines 61-69, the authors have employed terminologies that require further clarification, which is essential for a comprehensive understanding of this study. It is necessary for the authors to provide explicit definitions for key terms, specifically "safely managed sanitation services" and "access to basic and improved sanitation services." By doing so, readers will gain a clearer understanding of these concepts and their relevance to the study.
The authors have highlighted Malawi's initiatives to eradicate open defecation and have utilized a modeling approach to demonstrate the provision of sanitary facilities as a means to address this issue. In my personal experiences, I have observed that cultural factors play a significant role, as open defecation in open spaces is socially accepted and practiced despite the availability of latrines. Therefore, I would like to inquire whether the authors have considered incorporating such cultural factors into their modeling framework to comprehensively address the challenges associated with open defecation. If these cultural aspects cannot be incorporated into the model, it is important to acknowledge and address them as limitations of the study or potential areas for further research and improvement.
In lines 112-114, the statement mentions that the efforts have involved providing sanitary facilities to a population that previously lacked such provisions and also supplying new sanitary provisions to accommodate a growing population. However, it is not explicitly clear how "sanitary facilities" differ from "new sanitary provision" in this context. To ensure clarity, it would be helpful for the authors to provide a precise definition or explanation of these terms within the study.
In line 131, there appears to be a period (.) that is out of place.
In lines 333-334, it is suggested that the visibility or scale of Figure 4 could be enhanced to improve its readability. Additionally, Figure 5 could benefit from improvements to enhance its clarity.
The authors mentioned that in order to attain the sanitation goals, Malawi will need to accelerate the rate of sanitary provision. It is recommended that the authors explicitly discuss the specific aspects derived from their analysis, outlining the necessary actions that must be taken if the goals are to be achieved by 2030.
The subheadings "2.3 Status of access to adequate sanitation in 2018/19" and "3.1 Status of access to adequate sanitation in 2018/19" are identical, which can lead to confusion. It would be beneficial to rephrase these subheadings in the findings section to clearly indicate that they present results rather than duplicating the description of the status.
The paper does not encompass all the indicators utilized for measuring progress towards the goal, such as the percentage of the population with handwashing facilities. Consequently, the topic "Is SDG 6.2 achievable?" might be overly broad and potentially provocative. I would recommend that the authors refine the focus of the topic. Alternatively, the authors could review the indicators of SDG 6, develop a comprehensive framework, and demonstrate the connections between analysis indicators/data and SDG 6.2. In instances where certain indicators cannot be considered as references for analysis, the authors should explicitly state that they have not taken those particular indicators into account. This approach will enhance clarity and transparency in the study's methodology.
I kindly ask the authors, if feasible, to conduct a household survey that specifically addresses the behavioral and cultural aspects related to open defecation and latrine usage practices among the sampled families or communities. Incorporating this analysis into the revised version of the study would be beneficial as it would provide valuable insights and complement the existing analysis.
None
Author Response
Thank you so much for your detailed comments and your thoughts reading the paper, they are greatly appreciated. We have made the following changes to address your feedback.
- In line 37-39 we clarified the impact of waterborne disease and child deaths with providing a more updated, annual, global estimate of child deaths from diarrhoeal disease alone rather than the previous daily estimate. We hope that this makes it clearer.
- We have added additional definitions of both safely managed sanitation services as well as basic sanitation services alongside the preexisting improved sanitation definition.
- Thank you very much for your comment regarding the cultural aspects of open defecation, this is certainly well noted. Unfortunately, it was beyond the scope of this study to incorporate such cultural aspects as the study focused on a rather broad scoping evaluation of the level of open defecation across the country. We have added that the lack of cultural considerations surrounding open defecation is indeed a limitation to the discussion.
- We had used the term ‘new sanitary facilities’ to emphasise that the sanitary facilities were provided to a group who previously did not have sanitary facilities, however we recognise that this may have been confusing and the terminology ‘new’ has been removed for clarification. In the case of ‘new’ users this has been replaced with ‘additional’.
- To provide greater insight into the steps necessary for Malawi to achieve the goals of ending open defecation we have added a policy-focussed section to the discussion. This is specifically focussing on encouragement the revision of the 2006 sanitation policy which we believe will be critical in Malawi meeting these targets and was mentioned in personal correspondence with stakeholders.
- We have adjusted the subheadings to make it clear what is in the results section.
- We agree that there are many more aspects to Malawi achieving SDG6.2 than ending open defecation alone, however, without doing so open defecation will be a barrier to the Malawi achieving SDG6.2. For this reason, we felt that this title was appropriate. To add further clarification we have made reference to the specific target sdg6.2.1.a which directly addresses the proportion of the population using safely managed sanitation services and also clarified that this research focuses on the ending open defecation component of SDG6.2.
- We agree that a study addressing the cultural and behavioural aspects of pit-latrine usage would be very valuable to addressing open defecation in Malawi. The survey presented here represents a broad sample of communities from across Malawi and the addition of cultural and behavioural evaluations were not within the scope of this study. Furthermore, the ethics of the study do not allow for follow up surveys of the communities to provide more information into these factors. We are, however, evaluating an alternative survey on the nature and challenges surrounding open defecation on a smaller scale to provide further insight into these questions. This survey however involved an alternative study design, methodology, study area and collaborators and was therefore deemed to not be a good fit within this paper.
Thank you again for your comments and feedback. We hope that this sufficiently addresses your concerns and greatly appreciate your time and care in reading and responding to the paper.
Round 2
Reviewer 1 Report
Accept in present form
Accept in present form
Reviewer 3 Report
Thanks for addressing my comments, and I agree with them. I recommend that the authors provide a more explicit definition of "business as usual" to ensure clarity for the readers.
The paper is well-written, and I believe it could benefit from some proofreading or minor editing to enhance its overall quality.